# BENCHMARKING FEDERATED LEARNING FOR SEMANTIC DATASETS: FEDERATED SCENE GRAPH GENERATION

## ABSTRACT

Federated learning (FL) has recently garnered attention as a decentralized training framework that enables the learning of deep models from locally distributed samples while keeping the data privacy. Built upon the framework, immense efforts have been made to establish FL benchmarks, which provide rigorous evaluation settings that aim to control data heterogeneity across clients. Prior efforts have mainly focused on handling relatively simple classification tasks, where each sample is annotated with a one-hot label, such as MNIST, CIFAR, LEAF benchmark, etc. However, little attention has been paid to demonstrating an FL benchmark that handles complicated semantics, where each sample encompasses diverse semantic information from multiple labels, such as Scene Graph Generation / Panoptic Scene Graph Generation (SGG/PSG) with objects, predicates, and relations between objects. Because the existing benchmark is designed to distribute data in a narrow view of a single semantic, e.g., a one-hot label, managing the complicated *semantic heterogeneity* across clients when formalizing FL benchmarks is non-trivial. In this paper, we propose a benchmark process to establish an FL benchmark with controllable semantic heterogeneity across clients: two key steps are i) data clustering with semantics and ii) data distributing via controllable semantic heterogeneity across clients. As a proof of concept, we first construct a federated SGG/PSG benchmark, which demonstrates the efficacy of the existing PSG methods in an FL setting with controllable semantic heterogeneity of scene graphs.

## 1 INTRODUCTION

Federated learning (FL) has drawn great attention as a key framework for enabling decentralized training of deep models from the distributed private data to numerous edge clients. To keep the distributed local data private, the FL framework communicates the model parameters between the clients and the server; the server is not allowed to access data samples of clients McMahan et al. (2017). This property of FL that preserves data privacy makes it more crucial when deep models for tasks with license- or privacy-sensitive data, such as clinical data from medical institutions, private information from electronic edge devices, licensed contents from providers, etc.

Along with the rapid advancement of the algorithmic development of FL, great efforts have been dedicated to constructing FL benchmarks that allow reliable and rigorous evaluations for the existing FL methods. The existing FL benchmarks mostly rely on the existing datasets, including MNIST Deng (2012), CIFAR Krizhevsky et al. (2019), Celeb Liu et al. (2015), Twitter Caldas et al. (2018), etc, then building upon them, researchers focus on devising a decentralized training setting with controllable factors, such as data heterogeneity across clients, number of clients, ratio of participating clients, number of the maximum communication rounds, etc.

Among the factors of FL settings, *data heterogeneity* works as the most crucial factor that obviously exhibits the efficacy of different FL algorithms; when the data distribution strongly deviates across clients, a federation of local models typically fails with drastic performance drops Li et al. (2019). To construct a controllable and rigorous benchmark process with data heterogeneity, researchers intentionally diversify the prior distribution of one-hot label of samples across clients via random

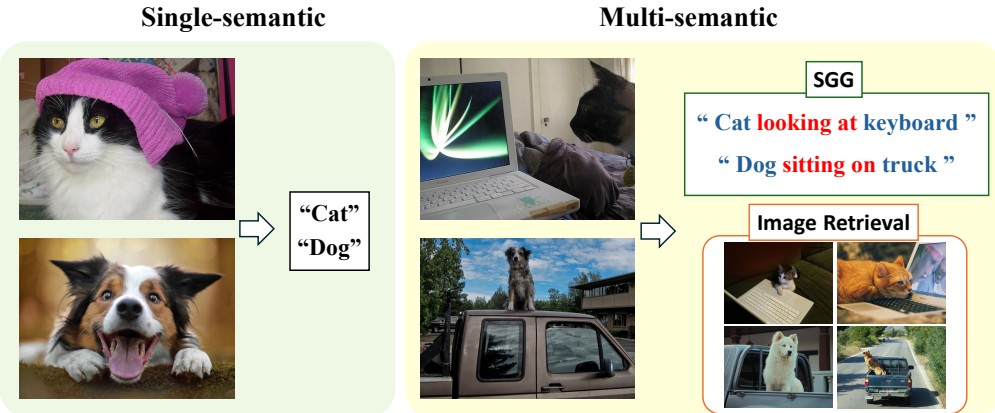

Figure 1: A task with a single semantic (left), e.g., an image classification task, utilizes the one semantic, i.e., the one-hot label of the image. Tasks with multiple semantics (right), e.g., image retrieval and Scene Graph Generation (SGG), leverage in-depth information, i.e., objects, predicates, and the relation between objects.

sampling of the priors from Dirichlet distribution Yurochkin et al. (2019); Wang et al. (2020a), or shard- or chunk-wise assignment of data McMahan et al. (2017).

Herein, we want to point out two key limitations of the existing FL benchmarks.

Firstly, the current benchmarks mostly handle relatively simple classification or regression tasks, where each sample is paired with a one-hot label or a single target value. However, deep training tasks currently become far beyond mere classification or recognition and consider highly complicated jobs to understand in-depth semantic information hidden in the given data sample, e.g., generating realistic samples Rombach et al. (2022); Ramesh et al. (2021), finding similar samples Gordo et al. (2017); Gordo & Larlus (2017); Radenović et al. (2018) from the users' prompts, or answering the queries about actions and objects shown in photos Antol et al. (2015); Alayrac et al. (2022). It is undoubtedly required to extend the current FL benchmark process to the tasks considering complicated semantics.

Secondary, there does not exist a task-agnostic FL benchmark process that devises controllable *semantic heterogeneity* of data across clients. In a simple image classification FL benchmark, for instance, the existing process focuses on a single semantic, i.e., the class label, and deviates the label distribution across clients to devise data heterogeneity (referring to the left image of Fig. 1). In contrast, when considering vision tasks, where each sample contains multiple semantics, devising data heterogeneity is non-trivial. As shown in the right image of Fig. 1, Scene Graph Generation (SGG), which is a core vision task for understanding the complicated semantics of a given image, a single sample bears multiple objects ('cat', 'keyboard', 'dog' and 'truck'), predicates ('looking at' and 'sitting on') and relations ('cat' → 'keyboard', 'dog' → 'truck'). Partitioning samples with such complex semantics into clients while controlling the semantic heterogeneity remains unexplored.

In this study, we propose an FL benchmark process that enables a evaluation of FL algorithms on multi-semantic datasets while controlling semantic heterogeneity. To break the aforementioned limitations, our process encompasses two key steps: i) discovering the semantic clusters by utilizing the collection of multiple annotations called 'category tensor'. ii) distributing data samples to multiple clients by considering the heterogeneity between the different semantic clusters.

As a proof of concept, we aim to construct the FL benchmark for the Panoptic Scene Graph Generation (PSG) task, which is one of the foundational tasks for computer vision research. We select SGG-related tasks because discovering scene graphs not only directly links to the fundamental understanding of visual perceptions but also works as a key module in bridging vision and language Chang et al. (2021). To the best of our knowledge, our work first attempts to establish the FL benchmarks for PSG and provides the evaluation results that demonstrate the effectiveness of the existing PSG baselines in decentralized training settings. The simulation results reveal that the methods tailored to tackle the long-tailed problem in PSG tasks, where some objects and predicates are more dominant than others, are shown to be robust in handling semantic heterogeneity in FL.

## 2 RELATED WORKS

### 2.1 FEDERATED LEARNING (FL) AND BENCHMARKS

FL has emerged as a pivotal framework for training deep learning models in a decentralized setting, enabling the preservation of clients' data privacy. This is achieved by ensuring that the data remains to each client, without the need to transfer sensitive information to a central server. Since the pioneering work of the Federated Averaging called FedAvg McMahan et al. (2017), which averages locally trained models, researchers mostly have been dedicated to handling the case with a strong heterogeneity of data across clients, resulting in the diverse strategies, including FedProx Li et al. (2020), SCAFFOLD Karimireddy et al. (2020), FedDyn Acar et al. (2021), FedSAM Qu et al. (2022); Caldarola et al. (2022), FedSMOO Sun et al. (2023), and FedGF Lee & Yoon (2024).

To establish thorough evaluation of the FL algorithms, a variety of **FL datasets** have been suggested for various tasks ranging from image classification Caldas et al. (2018); Li et al. (2022), natural language processing (NLP) Caldas et al. (2018); Lin et al. (2021), audio emotion recognition Zhang et al. (2023), multimodal learning Feng et al. (2023), to graph-based learning Wang et al. (2022). For organizing the agreed framework for the FL settings, several **FL testing environments** have been publicly researched and released, including Flower Beutel et al. (2020), FedML He et al. (2020), FedScale Lai et al. (2022), and FedLab Dun Zeng & Xu (2021).

Regardless of the datasets and environments, the most crucial factor in FL evaluation is demonstrating the effectiveness of methods with strong data heterogeneity across clients. When each sample contains a simple semantic, such as a single target label, there exist unified strategies to impose heterogeneity across clients by diversifying the prior distribution of the target label. Specifically, two main strategies include **i)** sampling the prior distribution of each client from Dirichlet distribution Acar et al. (2021), and **ii)** chucking per-class data samples into multiple shards, where a fixed number of shards are allocated to each client, yielding heterogeneity across clients Achituve et al. (2021); Lim et al. (2024).

However, when each sample bears multi-semantics, the current methods are not esaily extended, so many works rely on random split into clients, which cannot impose heterogeneity across clients. We suggest a benchmark process with well-controlled semantic heterogeneity across clients for fully evaluating deep models in FL settings with multi-semantics. One recent work has suggested a benchmark called FedNLP Lin et al. (2021) to impose semantic heterogeneity across clients, particularly in NLP field, but it relies on the pretrained language model to discover semantic clusters and cannot be extended to vision tasks. In contrast, our benchmark does not rely on extra pretrained models and considers the first-ever developed FL scene graph generation (SGG) testing environment.

### 2.2 PANOPTIC SCENE GRAPH GENERATION (PSG)

Scene graphs are crucial for scene understanding in computer vision tasks, representing objects (nodes) and predicates (relationships, edges) in a graph structure. Objects are commonly represented by bounding boxes. Predicting the bounding boxes and relationships between bounding boxes constitute scene graph generation (SGG). Recently, Conditional Random Field (CRF) Cong et al. (2018), TransE Zhang et al. (2017); Hung et al. (2020), CNN Woo et al. (2018); Yin et al. (2018), RNN/LSTM Xu et al. (2017); Zellers et al. (2018); Li et al. (2018); Tang et al. (2019); Wang et al. (2020b), GCN Herzig et al. (2018); Qi et al. (2019); Yang et al. (2018) based SGG algorithms were studied. Subsequently, the Panoptic Scene Graph Generation (PSG) task has been proposed Yang et al. (2022), which delves deeper into SGG by using panoptic segmentation masks instead of bounding boxes. The difference between PSG and SGG is that PSG uses panoptic segmentation Kirillov et al. (2019) masks rather than bounding boxes. Therefore, existing SGG models can be applied to PSG tasks. Research on PSG Zhou et al. (2023b;a); Li et al. (2024); Wang et al. (2023); Zhao et al. (2023) has been diverse and rapidly advancing in recent times.

**Long-tailed Problem:** The SGG/PSG tasks face the long-tailed problem Desai et al. (2021). Positional relationships among objects constitute the majority of the predicates, leading to a visual relationship complexity of $\mathcal{O}(N^2R)$ for $N$ objects and $R$ predicates Chang et al. (2021). This exacerbates the long-tailed problem in SGG/PSG data, prompting various recent approaches Lin et al. (2020); Tang et al. (2020); Yu et al. (2020); Abdelkarim et al. (2021); Chiou et al. (2021); Zhou et al. (2023a;b); Jin et al. (2023); Li et al. (2024) have been proposed to address this issue.

## 3 BACKGROUND

### 3.1 FEDERATED LEARNING

We briefly introduce the preliminaries of federated learning (FL) by focusing on the foundational baseline, i.e., FedAvg, McMahan et al. (2017). FL settings contain a single server and $K$ clients. The training process consists of iterative rounds, where the server and clients communicate the model parameters to each other. At each round $t$, a server initiates the round by broadcasting a global model $w^t$ to all clients. Each client then performs local training with its own data to obtain the locally trained model, i.e., $w_k^t$, where $k$ is the client index. After that, the server aggregates the locally trained model to compute the average model, which works as the global model of the next round:

$$w^{t+1} = \sum_{k=1}^{K} \frac{n_k}{n} w_k^t, \tag{1}$$

where $n_k$ is the number of data samples on client $k$, and $n$ is the total data samples across all clients. Notably, our formulation assumes full participation of all clients, but it can be extended to partial participation by letting a subset of clients participate in the aggregation for each round.

### 3.2 FORMALIZING HETEROGENEOUS SETTINGS

The heterogeneity of distributed data is a critical factor for FL. When data distribution across clients is homogeneously, then the data distribution is identical for all clients, the so-called independent and identically distributed (IID) case. Otherwise, when the distribution diversifies across clients, we call it the non-IID case. We here describe the existing methods for establishing the non-IID cases in FL.

**Label-based partition:** Label-based data partitioning is the most widely-used approach where the dataset is distributed according to the label of samples. For instance, shard-based partitioning McMahan et al. (2017) divides the dataset into shards with each shard containing data samples of one or a few classes. Each client receives one or more shards randomly, resulting in each client owning data samples that are biased toward specific classes. Dirichlet distribution-based partitioning Acar et al. (2021) offers a controllable and flexible method for simulating non-IID data distributions. The Dirichlet distribution is parameterized by a concentration parameter $\alpha$; as $\alpha$ gets close to 0, the sampled prior distribution is biased toward specific classes, in contrast, as $\alpha$ gets close to $\infty$, the sampled prior distribution tends to be uniform over all classes.

**Feature- or attribute-based partition:** A feature commonly refers to an attribute of data samples. For example, when predicting an outcome based on the values from specific sensors, the sensor's location can be its feature. In such cases, non-IID data can be constructed based on the sensor's location. The data features among clients may be completely non-overlapped, partial-overlapped, or full-overlapped Zhu et al. (2021). Non-overlapped features refer to cases where the features in the data are different from each other. For instance, some data might include 'gender' and 'age' as features, while other data might include 'height' and 'weight'. Partial-overlapped features refer to cases where the data shares some, but not all, features. For example, when taking photos of an object from the left and right sides, the images capture the same object but are not identical. Fully overlapped features refer to cases when the data features are completely identical, which is the most common scenario. Based on the degree of feature overlap, we can dynamically construct data heterogeneity.

**Temporal partition:** Temporal partitioning leverages the temporal and spatial variability of data to construct non-IID datasets. For instance, when utilizing stock market data, data heterogeneity can be created by assigning data from specific periods to individual clients.

In brief, feature-based and temporal partitioning are strongly related to the given task or datasets, so the applicability to a wide range of FL scenarios is severely limited. Therefore, most of the works with algorithmic developments of FL rely on the label-based partitioning method.

## 4 A BENCHMARK PROCESS FOR FL WITH MULTI-SEMANTIC DATASETS

For a given multi-semantic dataset, each data sample contains multiple annotations, i.e., $(x, \mathcal{Y}) \in \mathcal{D}$, where $x$ is an input, $\mathcal{Y} = \{y_1, \cdots, y_L\}$ is a multi-semantic label, $L$ is the possible number of labels

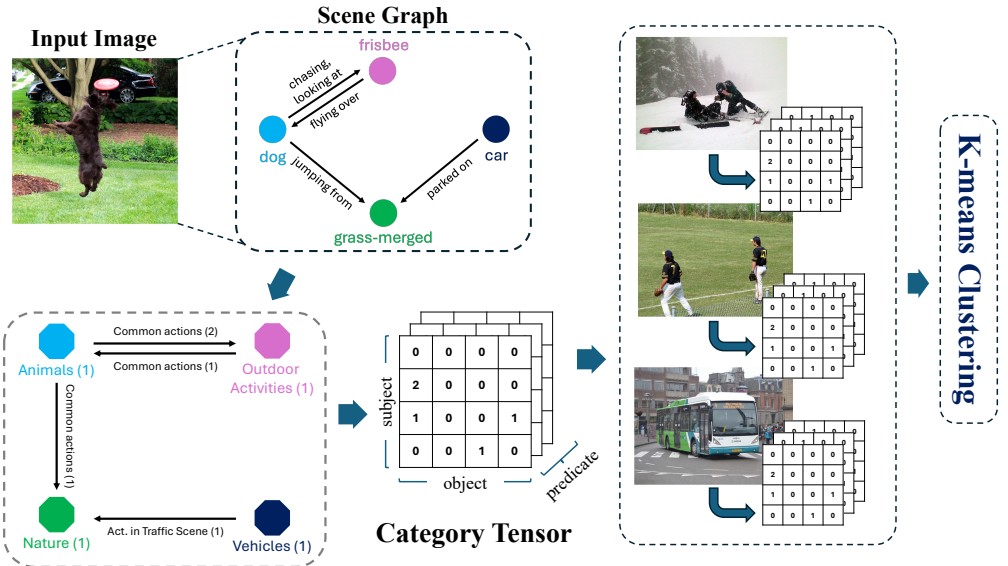

Figure 2: **Category Tensor K-means Clustering Pipeline**. The scene graphs of images are converted into category graphs, and these are further transformed into the category tensor. This process is applied to all image data, and K-means clustering is utilized to form clusters.

for each data sample, and $\mathcal{D}$ represents the dataset. Here, we introduce our benchmark process to distribute the multi-semantic data samples into $K$ multiple clients with controllable semantic heterogeneity. The key steps are twofold: i) discovering data clusters with different semantics and ii) data partitioning with controllable semantic heterogeneity across clients.

## 4.1 Discovering Data Clusters: $K$-means Clustering of Category Tensor

For a given multi-semantic $\mathcal{Y}$, we transform it to *category tensor* $\mathcal{F}$ by allocating each label $y_i$ into an orthogonal axis of the tensor, i.e., $\mathcal{F}(\mathcal{Y}) \in \mathbb{R}^{N_1 \times \cdots \times N_L}$, where there are $N_1, \cdots, N_L$ possible categories for each respective label of $\mathcal{Y}$. We then apply $K$-means Clustering on the collection of $\mathcal{F}(\mathcal{Y})_1^{|\mathcal{D}|}$ of overall dataset:

$$\mathcal{K}\Big(\mathcal{F}(\mathcal{Y})_1^{|\mathcal{D}|}\Big) \to \{\mathcal{C}_1, \cdots \mathcal{C}_n\}, \tag{2}$$

where $n$ is the discovered number of clusters and $\mathcal{C}_i$ indicates the collection of samples assigned to $i$-th cluster. With the obtained clustering, we can transform a data $(x, \mathcal{Y})$ into $(x, \mathcal{C})$ to impose the cluster label $\mathcal{C}$ with semantic information, which is a one-hot label with 1 for the assigned cluster. Through this clustering, we can perform the label-based partition while fully utilizing the multi-semantic information of each data sample with cluster label $\mathcal{C}$.

## 4.2 Data Partition with Semantic Heterogeneity

From Eq. (2), we have acquired $n$ clusters. It trivially raises the issue that the clusters are not evenly distributed, so the number of samples assigned to each cluster would deviate for different clusters, i.e., *cluster imbalance*. The *cluster imbalance* prevents rigorous evaluations of FL to handle semantic heterogeneity because a model becomes overfitted to dominant clusters without balanced training across different semantics. The cluster imbalance stems from the long-tailed problem, which is a key challenge in SGG task datasets. In other words, we have to create data heterogeneity for FL, which makes it difficult to distinguish from the long-tailed problem. If the amount of data in each cluster is equalized, the long-tailed problem may be partially alleviated. Furthermore, considering the federated learning scenario, this cluster imbalance is likely to bias the update of the global model in the update direction of users belonging to the dominant cluster. That is, it causes overfitting to a dominant cluster, which makes it difficult to closely compare each algorithm. Consequently, we need to equalize the data quantity of each cluster: $\hat{\mathcal{C}}_k = \text{Sample}(\mathcal{C}_k, m)$, for all $1 \leq k \leq n$, where

$m = \min_{k \in [n]}\{|\mathcal{C}_k|\}$, $|\mathcal{C}_k|$ is the cardinality of the $k$-th cluster $\mathcal{C}_k$, and $\mathrm{Sample}(\mathcal{C}_k, m)$ denotes a function that randomly selects $m$ data samples from cluster $\mathcal{C}_k$.

Now, we are ready to apply label-based partition to these clusters, enabling us to impose semantic heterogeneity. Our benchmark suggests two partition strategies as follows.

**Shard-based partition:** First, let each client choose $p(\leq n)$ clusters. We then split each cluster into disjoint shards or chunks, where the number of shards equals the number of clients who have selected the cluster. After splitting, the shards are distributed to the corresponding clients. If we set $p = n$, all clients are assigned to all clusters, making data distribution homogeneous. Otherwise, we can strengthen the heterogeneity by letting $p$ be small.

**Dirichlet distribution-based partition:** From the strategy suggested in Acar et al. (2021), the amount of data taken by each client from cluster $k$ is governed by the sampling from the Dirichlet distribution. We can design the non-IID data partition into $U$ clients by sampling $u$-th client's multinomial $\mathbf{p}_u \sim \mathrm{Dir}_n(\boldsymbol{\alpha})$ from Dirichlet distribution with $\boldsymbol{\alpha}$, where $\sum_{i=1}^n \mathbf{p}_{u,i} = 1$ and $\mathbf{p}_{u,i} \in [0,1]$ for all client $u \in \{1, ..., U\}$ and for all cluster $i \in \{1, ..., n\}$, and $\boldsymbol{\alpha} = (\alpha_1, \alpha_2, ..., \alpha_n)$ for all $\alpha_i \in (0, \infty]$ is a concentration parameter vector. Same as Acar et al. (2021), we set all the element of $\boldsymbol{\alpha}$ is same, $\alpha_1 = \alpha_2 = ... = \alpha_n$. Each client $u$ can sample training data from a dataset according to a proportion $\mathbf{p}_{u,i}$ without replacement for every cluster $i$. The heterogeneity of the data can be controlled by a value of $\alpha$. As $\alpha$ increases, the homogeneity of the data across clients increases, while as $\alpha$ decreases, the heterogeneity of the data across clients increases.

### 4.3 PROOF-OF-CONCEPT: FL BENCHMARK FOR PANOPTIC SCENE GRAPH GENERATION (PSG)

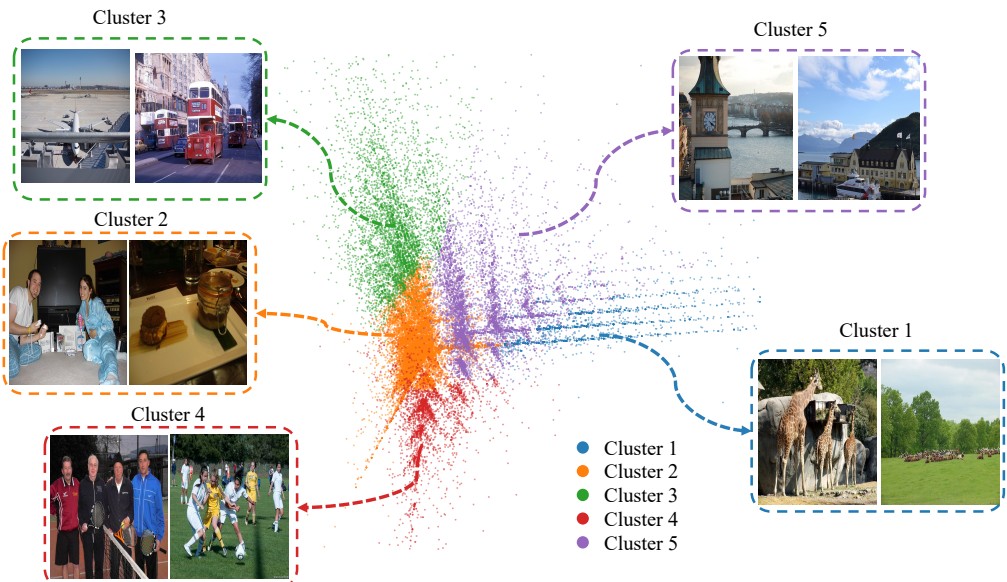

Figure 3: Semantic clusters of PSG dataset (a visualization via Principal Component Analysis)

Herein, we provide a proof-of-concept of our FL benchmark process by constructing the FL benchmark for PSG dataset Yang et al. (2022).

**i) Discovering data clusters:** PSG dataset contains object, subject, and predicate labels for each image sample. For simplicity, we utilize 13 object/subject categories and 7 predicate categories, which are the super-classes of fine-grained labels. Therefore, the dimension of the category tensor is $\mathcal{F}(\mathcal{Y}) \in \mathbb{R}^{13 \times 13 \times 7}$. For the category tensor, we perform $K$-means Clusters to obtain multiple semantic clusters. Consequently, we obtain 5 different clusters with discriminated semantics:

$$\mathcal{K}\left(\mathcal{F}(\mathcal{Y})_1^{|\mathcal{D}|}\right) \to \{\mathcal{C}_1, \mathcal{C}_2, \mathcal{C}_3, \mathcal{C}_4, \mathcal{C}_5\}. \tag{3}$$

Fig. 3 illustrates the sample images from each cluster. By examining the samples for each cluster, we observe the following distinguished features for each cluster and the imbalance between clusters:

- **Cluster 1** (occupying 5% of datasets): This cluster primarily contains images related to animals. We observe that it contains a large number of **animal objects** compared to others. The predicates are composed of actions that animals trivially perform.
- **Cluster 2** (occupying 58% of datasets): This cluster is dominated by **daily photographs of people**, which constitutes the largest portion of PSG dataset. This cluster mainly contains a significant number of objects that are related to daily activities by human beings, such as food photographs that frequently appear in daily life.
- **Cluster 3** (occupying 11% of datasets): This cluster mainly includes **urban landscape images and transportation-related photos**. Therefore, it encompasses many predicates related to vehicles, e.g., 'parking on' and 'driving (on).'
- **Cluster 4** (occupying 7% of datasets): This cluster is composed of images related to **sports**. It contains a large number of objects associated with sports and predicates such as 'playing' are more prevalent compared to others.
- **Cluster 5** (occupying 19% of datasets): This cluster corresponds to **urban/nature-combined landscapes**, which typically include buildings, the sky, and a river in the images. Due to objects related to natural elements, the predicates in this cluster are predominantly positional rather than action-oriented.

Notably, Clusters 2 and 4 contain somewhat similar images, mainly of 'people'. However, the predicates in Cluster 4 relate to sports activities, which is clearly discriminated from Cluster 2. Also, Cluster 3 and 5 look similar because of urban landscapes, but Cluster 3 leans to focus on cityscapes with transportation, and Cluster 5 focuses on urban/nature-combined city views. The qualitative visualization clearly demonstrates that our data clustering effectively and intuitively segments PSG dataset, leading to the splits given semantic information.

**ii) Data partition:** Based on the discovered 5 semantic clusters, our benchmark provides two options for data distribution: i) Shard-based partitioning and ii) Dirichlet distribution-based partitioning. Regardless of the choice of partitioning, when partitioning becomes heterogeneous, the data distribution at clients strongly deviates in the sense of semantic clusters, which leads to strong semantic heterogeneity. Otherwise, the data distribution of clients becomes homogeneous, yielding evenly distributed semantic information. The data partitioning is quite straightforward, so we will further describe the detailed settings in the following Section 5.

## 5 EXPERIMENTS: BENCHMARKS FOR PANOPTIC SGG IN FL

### 5.1 EXPERIMENT SETTINGS

We extensively evaluate the existing panoptic scene graph generation (PSG) models on our benchmark, including the following methods: IMP Xu et al. (2017), MOTIFS Zellers et al. (2018), VCTree Tang et al. (2019), and GPS-Net Lin et al. (2020).

**Dataset description:** For the detailed comparison of experiments, we utilize PSG dataset Yang et al. (2022) which includes diversified images with rich relational annotations, where each image is annotated with objects, panoptic segmentation masks, and fine-grained relationships between those objects. It not only identifies individual objects and their relationships but also includes stuff (amorphous background regions like "sky" or "grass"), which is often overlooked in other datasets. PSG dataset contains 49k images from COCO Lin et al. (2014) and Visual Genome Krishna et al. (2017). We provided detailed information on PSG dataset in Appendix A.

**Experiment setups:** We set up an FL scenario with one server and 100 clients, distributing the training data to each client. The test data for our benchmark was the same as PSG test dataset. Five active clients were randomly selected in each round, and the test data was evaluated using the aggregated global model from the server. Each client performs local training with one epoch and a batch size of 16. The total number of training rounds reaches up to 100, and we report the R/mR@K performance of the final averaged model. For specific details, such as the learning rate used for model training, please refer to the Appendix C.

Table 1: Comparison of the performance of PSG algorithms on the proposed FL benchmark

| R/mR @K | Algorithms | CL† | Random | Shard | | Dirichlet distribution | | |
|---|---|---|---|---|---|---|---|---|
| | | | | IID | non-IID | $\alpha = 10(\approx \text{IID})$ | $\alpha = 1$ | $\alpha = 0.2$ |
| R/mR @20 | IMP | 16.54 / 6.55 | 12.45 / 3.08 | 12.62 / 3.20 | 11.26 / 2.28 | 12.31 / 3.36 | 12.10 / 2.92 | 9.31 / 1.78 |
| | MOTIFS | 16.97 / 7.56 | 13.54 / 4.60 | 13.26 / 4.64 | 13.33 / 4.06 | 13.33 / 4.39 | 13.34 / 4.09 | 13.25 / 4.28 |
| | VCTree | 16.80 / 7.20 | 12.73 / 4.38 | 13.00 / 4.57 | 12.49 / 3.99 | 13.00 / 4.42 | 12.86 / 4.36 | 13.06 / 4.17 |
| | GPS-Net | 18.00 / 7.83 | 13.93 / 5.98 | 14.83 / 6.90 | 14.57 / 5.90 | 14.88 / 6.33 | 14.82 / 6.16 | 14.38 / 5.91 |
| R/mR @50 | IMP | 17.87 / 6.96 | 13.89 / 3.44 | 13.97 / 3.53 | 12.57 / 2.59 | 13.79 / 3.23 | 13.40 / 3.23 | 10.83 / 2.03 |
| | MOTIFS | 18.59 / 8.01 | 15.07 / 5.05 | 14.82 / 5.06 | 14.92 / 4.48 | 14.77 / 4.71 | 14.63 / 4.44 | 14.77 / 4.64 |
| | VCTree | 18.54 / 7.70 | 14.20 / 4.75 | 14.50 / 4.94 | 14.04 / 4.41 | 14.32 / 4.82 | 14.34 / 4.78 | 14.51 / 4.56 |
| | GPS-Net | 19.69 / 8.30 | 15.63 / 6.51 | 16.42 / 7.37 | 16.37 / 6.36 | 16.46 / 6.74 | 16.34 / 6.62 | 16.01 / 6.36 |
| R/mR @100 | IMP | 18.37 / 7.11 | 14.46 / 3.56 | 14.45 / 3.65 | 13.06 / 2.68 | 14.48 / 3.89 | 13.92 / 3.35 | 11.25 / 2.10 |
| | MOTIFS | 19.15 / 8.14 | 15.64 / 5.16 | 15.38 / 5.20 | 15.43 / 4.65 | 15.33 / 4.86 | 15.15 / 4.62 | 15.18 / 4.71 |
| | VCTree | 19.02 / 7.82 | 14.69 / 4.87 | 14.97 / 5.05 | 14.62 / 4.54 | 14.87 / 4.97 | 14.90 / 4.90 | 15.03 / 4.68 |
| | GPS-Net | 20.28 / 8.47 | 16.34 / 6.66 | 17.08 / 7.55 | 16.91 / 6.49 | 17.10 / 6.91 | 16.84 / 6.77 | 16.55 / 6.51 |

† For centralized learning (CL) is with a centralized dataset without considering the FL settings.
**Bold** refers the best performance and underline denotes the 2nd performance.

**Benchmark setups:** As described in Subsection 4.2, to ease the cluster imbalance, we randomly sampled data from each cluster to ensure an equal amount of data for each cluster. We tested 6 types of data partitioning as follows: **(1) Random**, where data is distributed randomly among all clients, ensuring nearly equal sizes for each. **(2) Shard-based partition IID**, where we set $p = 5$, where $p$ is the number of clusters that client sample from. As aforementioned, when $p$ equals the number of clusters, the data from each cluster is equally distributed among 100 clients, i.e., an IID case. **(3) Shard-based partition non-IID**, where we set $p = 1$ for imposing semantic heterogeneity. Each cluster is assigned 20 clients, and all clients have the same amount of data. **(4), (5) and (6) Dirichlet distribution-based partition** ranging from an IID case to a strong non-IID case, where we tested three different levels of semantic heterogeneity by using $\alpha = [10, 1, 0.2]$.

**Metrics:** By following the work of Zhou et al. (2023a) that has first suggested PSG task, we use 'Recall@K (R@K)' and 'mean Recall@K (mR@K)' as the performance metrics, which respectively calculate the triplet recall and mean recall for every predicate category, given the top K triplets from PSG model. $K$ varies from 20 to 100. Moreover, R@K is dominated by high-frequency relations, and mR@K assigns equal weight to all relation classes. In datasets with severe long-tailed problems, like PSG dataset, mR@K can provide more meaningful insights into model performance.

## 5.2 IN-DEPTH ANALYSIS

Our intuition is that the performance of models is expected to show the following order: Centralized learning (CL) $\geq$ IID $\geq$ Random $\geq$ non-IID, when our benchmark effectively imposes semantic heterogeneity for the FL setting. And, the experimental results follow our intuition well.

**Results:** Table 1 shows the test accuracy on the test set of PSG dataset. We have focused on the Mean Recall (denoted as 'mR') performance. Also, we focus on the most challenging case with $K = 20$. **i) CL vs. IID.** The performance has been mostly degraded when comparing CL and IID cases. The averaged gaps for mR@20 are $-2.45\%$ and $-2.71\%$, for 'Shard-IID' and 'Dir($\alpha = 10$)', respectively. Each client has approximately 114 images, and due to the limited data, there appears to be a performance difference between the CL and IID scenarios. CL can collectively form a mini-batch across clients, but IID forms a mini-batch per client in a decentralized manner. **ii) IID vs. Random.** When data is randomly divided, it will tend to have a distribution close to IID, so that there is a minimal performance drop. The averaged gaps for mR@20 are $-0.32\%$ and $-0.12\%$, for 'Shard-IID' and 'Dir($\alpha = 10$)', respectively. The results confirm that the random partitioning naively conducted in prior studies is not suitable for imposing semantic heterogeneity, showing similar results as the IID case. **iii) IID vs. non-ID.** We confirm large performance degradations in most cases. First, in the case of a shard-based partition, the averaged gap for mR@20 is $-0.77\%$. Second, in the case of the Dirichlet distribution-based partition, i.e., comparing Dir($\alpha = 10$) and Dir($\alpha = 0.2$), the averaged gap for mR@20 is shown to be $-0.64\%$. The performance drops from IID to non-IID reveal that PSG algorithms struggle to aggregate a global model when facing a string semantic heterogeneity. Also, when a PSG algorithm shows a minimal performance drop due to the heterogeneity, it directly demonstrates the robustness against semantic heterogeneity. MOTIFS shows the outliers in mR, where the moderate non-IID case ($\alpha = 1$) compared to the non-IID case ($\alpha = 0.2$) shows minimal differences: 4.09% vs. 4.28% in mR@20, and 4.62% vs. 4.71% in mR@100. Although the results may seem unexpected, the differences are not significant. Notably, when $\alpha = 10$, corresponding to

the IID case, shows the best performance: 4.39% in mR@20 and 4.86% in mR@100, aligning with our expectations. Based on this observation, we conjecture that the behavior at the moderate non-IID can be a little shaky in a few cases, but it behaves as expected in the IID case.

**PSG Model comparisons:** We here to discuss the robustness of the existing PSG algorithms against semantic heterogeneity. We conclude that IMP Xu et al. (2017) is shown to be relatively vulnerable in handling semantic heterogeneity in FL, i.e., a large gap of $-1.58\%$ for mR@20 is observed when comparing Dir($\alpha = 10$) and Dir($\alpha = 0.2$). Compared to others, it has a relatively smaller model architecture and suffers from the long-tailed problem in PSG dataset. We conjecture that the aspects of IMP lead to considerable performance drops in our non-IID testing. VCTree includes a tree construction process trained through reinforcement learning, resulting in a more complex model structure compared to MOTIFS. Consequently, in a FL scenario with small-scale client data, VCTree's performance degraded. GPS-Net employs key elements, e.g., DMP, NPS, ARM, to resolve the long-tailed problem. We conjecture that it yields the outperforming results of GPS-Net in our FL benchmarks. In FL, clients have a small number of data samples, which makes worse the long-tailed problem. Because GPS-Net has two key factors that pay more attention to objects and predicates with smaller occurrences, it shows remarkable performances.

## 5.3 FL ALGORITHM: FEDAVGM

From Table 1, we confirmed that the improvements in PSG algorithms remain effective in the FL scenario. Subsequently, it is necessary to verify whether the improvements in FL algorithms are also valid within our benchmark. We present the result of applying FedAvgM Hsu et al. (2019) in Table 2 where it utilizes the momentum in updating a global model on the server side and relieve the largely varying directions of local update due to the stochastic variance across clients. A description of the training of FedAvgM is as follows: FedAvgM updates the global model, i.e., $w_g^{r+1} = w_g^r - v^r$ where $v^r = \beta v^{r-1} + \sum_{k=1}^{K} \frac{n_k}{n} \Delta w_k^r$, $\beta$ is the momentum hyperparameter for FedAvgM, $n_k$ is the number of examples, $\Delta w_k^r$ is the weight update from $k$'s client, and $n = \sum_{k=1}^{K} n_k$.

**Results:** FedAvgM sufficiently improves the performance of all algorithms. For R/mR@100, there is an average performance improvement of +0.93% in the Shard-IID case, and +1.30% in the Shard-nonIID case. This algorithm is proposed to address the data heterogeneity, and the performance improvements observed through its application support the validity of our approach, indicating that it effectively constructs federated learning scenarios. The performance of GPS-Net in the Shard non-IID case shows a negligible gap (i.e., $\leq 0.03\%$), indicating that GPS-Net already incorporates factors that mitigate the effects of heterogeneity. Thus, it is minimally influenced by FedAvgM.

Table 2: Comparison of the FedAvg and FedAvgM performances of PSG algorithms.

| R/mR@K | Method | FedAvg | | FedAvgM | |
|---|---|---|---|---|---|
| | | Shard IID | Shard non-IID | Shard IID | Shard non-IID |
| R/mR@20 | IMP | 12.62 / 3.20 | 11.26 / 2.28 | 13.33 / 3.48 (+0.28%) | 13.23 / 3.83 (+1.55%) |
| | MOTIFS | 13.26 / 4.64 | 13.33 / 4.06 | 15.63 / 6.04 (+1.40%) | 15.47 / 5.80 (**+1.74%**) |
| | VCTree | 13.00 / 4.57 | 12.49 / 3.99 | 15.45 / 6.14 (**+1.57%**) | 15.39 / 5.66 (+1.67%) |
| | GPS-Net | **14.83 / 6.90** | **14.57 / 5.90** | **16.66 / 7.21** (+0.31%) | **16.18 / 5.91** (+0.01%) |
| R/mR@50 | IMP | 13.97 / 3.53 | 12.57 / 2.59 | 14.63 / 3.87 (+0.34%) | 14.73 / 4.24 (+1.65%) |
| | MOTIFS | 14.82 / 5.06 | 14.92 / 4.48 | 17.11 / 6.41 (+1.35%) | 17.23 / 6.23 (**+1.75%**) |
| | VCTree | 14.50 / 4.94 | 14.04 / 4.41 | 17.21 / 6.59 (**+1.65%**) | 17.02 / 6.10 (+1.69%) |
| | GPS-Net | **16.42 / 7.37** | **16.37 / 6.36** | **18.47 / 7.76** (+0.39%) | **18.00 / 6.33** (-0.03%) |
| R/mR@100 | IMP | 14.45 / 3.65 | 13.06 / 2.68 | 15.09 / 3.97 (+0.32%) | 15.32 / 4.38 (+1.70%) |
| | MOTIFS | 15.38 / 5.20 | 15.43 / 4.65 | 17.65 / 6.55 (+1.35%) | 17.83 / 6.38 (**+1.73%**) |
| | VCTree | 14.97 / 5.05 | 14.62 / 4.54 | 17.75 / 6.74 (**+2.09%**) | 17.58 / 6.30 (**+1.76%**) |
| | GPS-Net | **17.08 / 7.55** | **16.91 / 6.49** | **18.94 / 7.90** (+0.35%) | **18.68 / 6.52** (-0.03%) |

($\cdot$) indicates the difference in mR@K when the FedAvgM algorithm is applied compared to FedAvg.

## 5.4 VARIOUS FL SCENARIOS

In this section, we conduct a series of experiments to investigate the impact of different factors on federated learning performance. Federated learning operates in diverse environments, making it essential to test various scenarios to better understand how the approach performs. By manipulating key parameters such as the total number of clients, participation rates, and federated learning

Table 3: Comparison of performances for the number of total clients.

| R/mR@100 | Method | Shard IID | Shard non-IID |
|---|---|---|---|
| Clients 50 | IMP | 15.62 / 4.60 | 14.92 / 4.70 |
| | MOTIFS | 18.24 / **7.74** | **18.22** / 6.88 |
| | VCTree | 16.61 / 6.51 | 16.89 / 6.40 |
| | GPS-Net | **18.54** / 7.22 | 18.11 / **7.29** |
| Clients 100 | IMP | 14.45 / 3.65 | 13.06 / 2.68 |
| | MOTIFS | 15.38 / 5.20 | 15.43 / 4.65 |
| | VCTree | 14.97 / 5.05 | 14.62 / 4.54 |
| | GPS-Net | **17.08** / **7.55** | **16.91** / **6.49** |
| Clients 200 | IMP | 12.80 / 3.00 | 12.37 / 2.30 |
| | MOTIFS | **15.22** / **5.23** | **15.43** / 5.08 |
| | VCTree | 12.54 / 3.56 | 12.4 / 3.33 |
| | GPS-Net | 13.69 / 5.22 | 13.49 / **5.09** |

Table 4: Comparison of performances for the number of participation rates.

| R/mR@100 | Method | Shard IID | Shard non-IID |
|---|---|---|---|
| # of clients 5 | IMP | 14.45 / 3.65 | 13.06 / 2.68 |
| | MOTIFS | 15.38 / 5.20 | 15.43 / 4.65 |
| | VCTree | 14.97 / 5.05 | 14.62 / 4.54 |
| | GPS-Net | **17.08** / **7.55** | **16.91** / **6.49** |
| # of clients 20 | IMP | 13.91 / 3.25 | 15.31 / 4.04 |
| | MOTIFS | **17.3** / **6.39** | **16.83** / **6.23** |
| | VCTree | 15.03 / 4.83 | 14.92 / 4.73 |
| | GPS-Net | 16.81 / 6.36 | 16.66 / 6.04 |

algorithms, we provide a comprehensive analysis of their effects on overall performance. For additional results, please refer to the Appendix B.

**Total clients:** We evaluate performances to examine the effect of number of total clients, i.e., 50, 100, and 200, in Table 3. The 50 clients case has twice as much data per client compared to the 100 clients case. As the number of data samples increases, meaning the total number of clients decreases, performance improves. Notably, VCTree shows a $2\times$ larger mR@K for 50 clients compared to 200 clients in the Shard non-IID case, indicating that VCTree is highly sensitive to the number of data samples, which is a critical factor in FL settings. In contrast, both MOTIFS and GPS-Net are less affected by the number of data samples, with GPS-Net achieving an mR@K greater than 5.09 and MOTIFS exceeding 4.65, significantly outperforming both IMP and VCTree across all cases.

**Participation rates:** We also evaluate the performance according to the number of participating clients, i.e., 5 and 20, in Table 4. As the number of participants increased, the performance of MOTIFS improved remarkably. According to previous FL studies, mainly handled the image classification task, increasing participation rate leads to improvement of performance in FL environments. However, in PSG task, as the number of users increases, the performance does not show similar behaviors without MOTIFS. In other words, rather than increasing the number of participants in each round, the larger amount of data of each participant can expect greater performance improvement in this task.

Additionally, we have to focus on the performances of IMP in various FL scenarios. IMP is the oldest algorithm in our experiments and shows a lower performance in the experiments of existing studies Tang et al. (2019); Yang et al. (2022). Therefore, the performance seems to be poor before being affected by data heterogeneity, making a detailed comparison difficult.

# 6 CONCLUSION

This study introduces an innovative benchmark for evaluating federated learning (FL) algorithms on complex semantic datasets in the vision domain such as SGG/PSG tasks. Our proposed benchmark process addresses the challenge of managing semantic heterogeneity across clients by employing semantic-based data clustering and controlled data partitioning, thereby providing first-ever test benchmarks for SSG/PSG in the FL setting. Extensive experimental results demonstrate the effectiveness of our benchmark in capturing real-world data distributions and providing meaningful insights into the performance of various PSG algorithms within FL scenarios.

# 7 IMPACT STATEMENTS

Our findings reveal that data heterogeneity influences model performance, showing that performance tends to decrease as heterogeneity increases. These results underscore the importance of designing FL benchmarks that accurately reflect the complexities inherent in real-world data. By offering a structured framework and benchmark for FL with multiple semantics, this study lays the groundwork for future research in federated (panoptic) scene graph generation and related domains. The benchmark and accompanying code are made publicly available to the research community, fostering transparency and reproducibility in FL research. We expect that this research will aid in the extension of various vision tasks to FL. Future research may explore additional strategies to mitigate the adverse effects of data heterogeneity and extend our benchmark to other complex semantic tasks.

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

## A  DATASET

PSG dataset leverages VG150 Krishna et al. (2017) and COCO Lin et al. (2014) datasets, which are not perfect but popularly used in SGG tasks, by integrating their comprehensive object and relationship annotations into a more focused dataset designed specifically for the scene graph generation. To be specific, PSG dataset directly inherits the panoptic segmentation annotations from COCO. On the other hand, the VG150 dataset contains many 'trivial' (not meaningful) predicates with the direction of predicates (e.g., of in hair-'of'-man, has in man-'has'-head), and PSG dataset gets rid of these predicates. PSG dataset contains 133 objects and 374 relationships, sufficiently covering the diversity from the VG150 and COCO datasets while compensating for their limitations. These relationships are more detailed and extensive than those in other datasets, allowing for richer scene graph representations.

The training dataset consists of 46,563 images, while the test dataset is composed of the remaining data. Table 5 describes the information about the dataset we have composed with Category Tensor K-means Clustering. An imbalanced dataset refers to a dataset where the number of data points in each cluster is not equal after clustering. As mentioned in Sec 4.2, to eliminate cluster imbalance, we randomly sampled the data from each cluster to match the quantity of the smallest cluster, Cluster 0. This process ensured that all clusters had the same amount of data, resulting in a balanced dataset.

Table 5: Dataset Information

| Data | Amount | |
|---|---|---|
| PSG dataset - Train | 46 K | |
| **Cluster** | **Imbalanced (100%)** | **Balanced (24.5%)** |
| Cluster 0 | 2.2K (4.9%) | 2.2K |
| Cluster 1 | 27K (58.1%) | 2.2K |
| Cluster 2 | 5.1K (10.9%) | 2.2K |
| Cluster 3 | 3.3K (7.1%) | 2.2K |
| Cluster 4 | 8.8K (19.0%) | 2.2K |

### A.1  CATEGORIES FOR PSG DATASET

The category lists we used for generating the category tensor are Table 6 for Subject/Object and Table 7 for Predicate. We should structure the categories as shown in these tables, depending on the dataset we are applying it to and the objects and predicates involved.

## B  ADDITIONAL RESULTS

### B.1  TOTAL CLIENTS

We provide varying total client experiment results in Table 8.

### B.2  PARCIPATION RATES

We provide varying participation rates experiment results in Table 9.

## C  CONFIGURATION

We set up an FL scenario with one server and 100 clients, distributing the training data of the existing PSG dataset Yang et al. (2022) to the 100 clients. The test data for our benchmark was the same as PSG test data. In each round, five active clients were randomly selected, and the test data was evaluated using the aggregated global model from the server. Each client performs local training with one epoch and a batch size 16. The total number of training rounds reaches up to 100, and we report the R/mR@K performance of the final averaged model. Following the benchmark in Li et al. (2024), we set the SGD optimizer with a learning rate of 0.02, momentum of 0.9, weight decay of 0.0001, and gradient clipping with a max L2 norm of 35.

Table 6: Subject/Object Categories

| Categories | Subject/Object |
|---|---|
| Person (1) | person |
| Vehicles (8) | bicycle, car, motorcycle, airplane, bus, train, truck, boat |
| Road Objects (5) | banner, traffic light, fire hydrant, stop sign, parking meter |
| Furniture (7) | bench, chair, couch, potted plant, bed, dining table, rug-merged |
| Animals (10) | bird, cat, dog, horse, sheep, cow, elephant, bear, zebra, giraffe |
| Clothing and Accessories (5) | backpack, umbrella, handbag, tie, suitcase |
| Outdoor Activities (12) | frisbee, skis, snowboard, sports ball, kite, baseball bat, baseball glove, skateboard, surfboard, tennis racket, playingfield, net |
| Kitchen and Dining (7) | bottle, wine glass, cup, fork, knife, spoon, bowl |
| Food (11) | banana, apple, sandwich, orange, broccoli, carrot, hot dog, pizza, donut, cake, food-other-merged |
| Household Items (23) | curtain, blanket, toilet, tv, laptop, mouse, remote, keyboard, cell phone, microwave, oven, toaster, sink, refrigerator, book, pillow, towel, clock, vase, scissors, teddy bear, hair drier, toothbrush |
| Structures (20) | bridge, house, tent, door-stuff, wall-other-merged, building-other-merged, pavement-merged, ceiling-merged, wall-brick, wall-stone, wall-tile, wall-wood, stairs, railroad, road, roof, floor-wood, platform, floor-other-merged, fence-merged |
| Nature (14) | flower, fruit, gravel, river, sea, tree-merged, snow, sand, water-other, mountain-merged, grass-merged, dirt-merged, rock-merged, sky-other-merged |
| Misc. (10) | cardboard, counter, light, mirror-stuff, shelf, window-blind, window-other, cabinet-merged, table-merged, paper-merged |

## C.1 HARDWARE CONFIGURATION

We conducted our experiments on a server with an NVIDIA A5000 GPU, Intel Xeon Gold processors, and 256 GB RAM. When running a single trial of training with 1 A5000 GPU, it takes 10 hours for the convergence.

Table 7: Predicate Categories

| Categories | Predicate |
|---|---|
| Positional Relations (6) | over, in front of, beside, on, in, attached to |
| Common Object-Object Relations (5) | hanging from, on back of, falling off, going down, painted on |
| Common Actions (31) | walking on, running on, crossing, standing on, lying on, sitting on, leaning on, flying over, jumping over, jumping from, wearing, holding, carrying, looking at, guiding, kissing, eating, drinking, feeding, biting, catching, picking, playing with, chasing, climbing, cleaning, playing, touching, pushing, pulling, opening |
| Human Actions (4) | cooking, talking to, throwing, slicing |
| Actions in Traffic Scene (4) | driving, riding, parked on, driving on |
| Actions in Sports Scene (3) | about to hit, kicking, swinging |
| Interaction between Background (3) | entering, exiting, enclosing |

Table 8: Comparison of the performance of PSG algorithms to observe impact of varying total client. The total amount of training data is the same. The training data is distributed to each client.

| R/mR@K | Method | Total client 50 | | Total client 100* | | Total client 200 | |
|---|---|---|---|---|---|---|---|
| | | Shard IID | Shard non-IID | Shard IID | Shard non-IID | Shard IID | Shard non-IID |
| R/mR@20 | IMP | 13.56 / 4.0 | 12.89 / 4.2 | 12.62 / 3.20 | 11.26 / 2.28 | 10.91 / 2.6 | 10.38 / 2.0 |
| | MOTIFS | 16.1 / 7.11 | 16.13 / 6.15 | 13.26 / 4.64 | 13.33 / 4.06 | 13.19 / 4.7 | 13.45 / 4.54 |
| | VCTree | 14.59 / 5.94 | 14.75 / 5.87 | 13.00 / 4.57 | 12.49 / 3.99 | 10.53 / 3.04 | 10.42 / 2.83 |
| | GPS-Net | 16.44 / 6.67 | 15.97 / 6.74 | 14.83 / 6.90 | 14.57 / 5.90 | 11.63 / 4.68 | 11.31 / 4.42 |
| R/mR@50 | IMP | 15.10 / 4.4 | 14.33 / 4.6 | 13.97 / 3.53 | 12.57 / 2.59 | 12.23 / 2.9 | 11.89 / 2.2 |
| | MOTIFS | 17.76 / 7.6 | 17.65 / 6.71 | 14.82 / 5.06 | 14.92 / 4.48 | 14.67 / 5.09 | 14.89 / 4.96 |
| | VCTree | 16.09 / 6.39 | 16.34 / 6.23 | 14.50 / 4.94 | 14.04 / 4.41 | 10.42 / 2.83 | 11.91 / 3.21 |
| | GPS-Net | 17.97 / 7.06 | 17.55 / 7.15 | 16.42 / 7.37 | 16.37 / 6.36 | 13.13 / 5.09 | 12.94 / 4.93 |
| R/mR@100 | IMP | 15.62 / 4.6 | 14.92 / 4.7 | 14.45 / 3.65 | 13.06 / 2.68 | 12.80 / 3.0 | 12.37 / 2.3 |
| | MOTIFS | 18.24 / 7.74 | 18.22 / 6.88 | 15.38 / 5.20 | 15.43 / 4.65 | 15.22 / 5.23 | 15.43 / 5.08 |
| | VCTree | 16.61 / 6.51 | 16.89 / 6.4 | 14.97 / 5.05 | 14.62 / 4.54 | 12.54 / 3.56 | 12.4 / 3.33 |
| | GPS-Net | 18.54 / 7.22 | 18.11 / 7.29 | 17.08 / 7.55 | 16.91 / 6.49 | 13.69 / 5.22 | 13.49 / 5.09 |

**Numbers are borrowed from the Table 1.

## C.2 SOFTWARE ENVIRONMENT

- Operating System: Ubuntu 22.04.3 LTS
- Deep Learning Framework: PyTorch 1.7.0
- Other Dependencies: CUDA 10.1, Python 3.7

## D CONVERGENCE BEHAVIOR

We present the convergence behaviors of four models on the shard and Dirichlet distribution based partition method in Fig. 4.

When we compare the non-IID and IID cases of shard, GPS-Net shows remarkable performance improvement in IID. GPS-Net Lin et al. (2020) has three key modules (DMP, NPS, ARM). In previous study Yang et al. (2022), DMP was the key module for the high performance in VG, which has the direction of predicates in the dataset (e.g., of in hair-of-man, has in man-has-head). But in PSG dataset, they removes these predicates, the DMP module has a lower effect on performance.

Table 9: Comparison of the performance of PSG algorithms to observe impact of varying participation rates. The total number of clients is 100, and when the partition rate is 20, twenty clients are selected in each round.

| R/mR@K | Method | participation rate 5* | | participation rate 20 | |
|---|---|---|---|---|---|
| | | Shard IID | Shard non-IID | Shard IID | Shard non-IID |
| R/mR@20 | MOTIFS | 13.26 / 4.64 | 13.33 / 4.06 | 15.04 / 5.73 | 14.94 / 5.71 |
| | VCTree | 13.00 / 4.57 | 12.49 / 3.99 | 12.94 / 4.29 | 12.96 / 4.22 |
| | GPS-Net | 14.83 / 6.90 | 14.57 / 5.90 | 14.81 / 5.83 | 14.58 / 5.51 |
| R/mR@50 | MOTIFS | 14.82 / 5.06 | 14.92 / 4.48 | 16.7 / 6.23 | 16.37 / 6.08 |
| | VCTree | 14.50 / 4.94 | 14.04 / 4.41 | 14.55 / 4.7 | 14.33 / 4.56 |
| | GPS-Net | 16.42 / 7.37 | 16.37 / 6.36 | 16.34 / 6.25 | 16.02 / 5.86 |
| R/mR@100 | MOTIFS | 15.38 / 5.20 | 15.43 / 4.65 | 17.3 / 6.39 | 16.83 / 6.23 |
| | VCTree | 14.97 / 5.05 | 14.62 / 4.54 | 15.03 / 4.83 | 14.92 / 4.73 |
| | GPS-Net | 17.08 / 7.55 | 16.91 / 6.49 | 16.81 / 6.36 | 16.66 / 6.04 |

**Numbers are borrowed from the Table 1.

However, we conjecture that other modules (NPS, ARM) that designed to solve long-tailed problems are effective in the FL scenarios. As a result, the performance of GPS-Net improved the fastest. However, GPS-Net showed a decrease in convergence speed under the non-IID situation. MOTIFS Zellers et al. (2018) and VCTree Tang et al. (2019) show similar behaviors in IID and non-IID cases, and they also have the same model structures(LSTM) and do not consider the long-tailed problem. These two algorithms do not seem to be significantly affected in terms of convergence speed by the non-IID situation. IMP Xu et al. (2017) shows poor performance compared to others because of message passing that relies on the direction of predicates, which does not come out in PSG dataset. And IMP also showed a decrease in convergence speed under the non-IID situation.

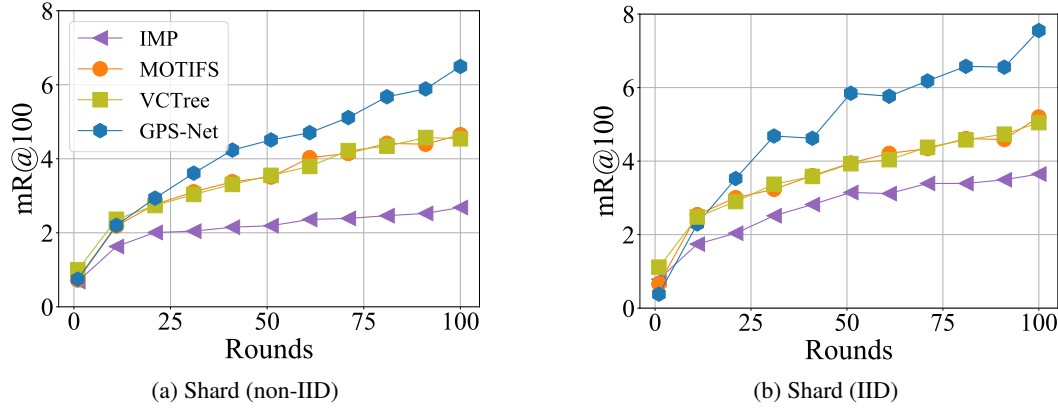

(a) Shard (non-IID)                    (b) Shard (IID)

Figure 4: Convergence behaviors of the balanced dataset for IID and non-IID cases

We also show the convergence behavior for Dirichlet distribution in Fig. 5, where $\alpha$ is the $[0.2, 1, 10]$. It shows similar results to behaviors of the shard-based partition scenario. Overall, as data heterogeneity increased, there was a decrease in performance. However, MOTIFS and VCTREE still do not seem to be significantly affected in terms of convergence speed.

# E   ANALYSIS OF COMMUNICATION COST

Communication efficiency is one of the most important factors because communication costs are very crucial in a practical FL scenario. When thinking of communication 'rate', as in a scenario with limited communication resources, we compare the algorithms by rigorously measuring the actual communication costs required to reach the same level of performance.

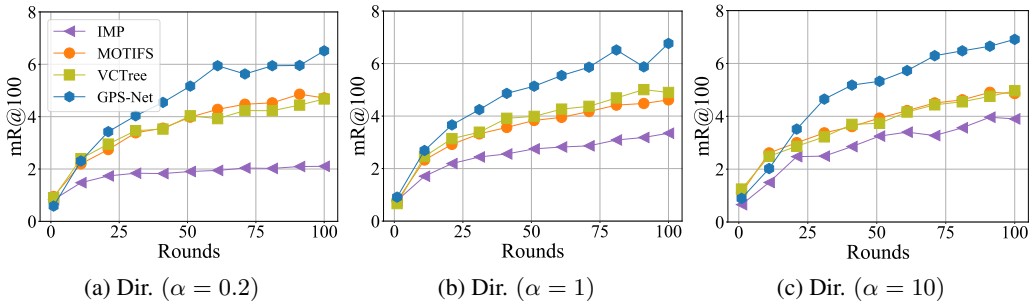

(a) Dir. ($\alpha = 0.2$)    (b) Dir. ($\alpha = 1$)    (c) Dir. ($\alpha = 10$)

Figure 5: Convergence behaviors of the balanced dataset for Dirichlet distribution

In fact, we have shown the convergence plots of each algorithm in the Fig. 4 and 5. By revisiting the results, we have computed the required communication costs to precisely compare the communication efficiency of each algorithm.

Specifically, we calculate the total communication cost, i.e., the number of model parameters multiplied by the communication round, to reach the 'mR@100 = 2' (the reason for the target performance is that the IMP algorithm shows the worst performance and converges near 2). We show the number of model parameters for each algorithm and the resulting communication costs required in Table 10.

IMP has the smallest number of model parameters but the highest communication cost. In contrast, GPS-Net has a similar number of model parameters with IMP, which accounts for half of the total communication cost, denoting that GPS-Net is the resource-efficient scene graph generation method in FL. GPS-Net shows remarkable performance because it has key elements, e.g., DMP, NPS, and ARM, to resolve the long-tailed problem. We conjecture that is why it shows the rapid convergence behavior to higher accuracies with fewer communication costs.

Table 10: Comparison of Communication Cost

| Method | # of model parameters | Shard IID | Shard non-IID |
|--------|----------------------|-----------|---------------|
| IMP | 32M | 64(x 1) | 64(x 1) |
| MOTIFS | 63M | 63(x 0.98) | 63(x 0.98) |
| VCTree | 59M | 59(x 0.92) | 59(x 0.92) |
| GPS-Net | 37M | 37(x 0.57) | 37(x 0.57) |

# F    CLUSTER IMBALANCE EFFECT

Table 11: Comparison of SGG FL performances (R/mR@K) with FL-PSG Dataset

| R/mR@K | Algorithms | CL Yang et al. (2022) | Random | Shard IID | Shard non-IID |
|--------|-----------|----------------------|--------|-----------|---------------|
| R/mR @20 | IMP | 16.5 / 6.52 | 16.10 / 5.68 | 16.38 / 5.97 | 15.50 / 4.75 |
| | MOTIFS | 20.0 / 9.10 | 16.66 / 6.52 | 16.64 / 6.60 | 16.34 / 6.32 |
| | VCTree | 20.6 / 9.70 | 16.49 / 6.42 | 16.93 / 7.03 | 16.46 / 6.09 |
| | GPS-Net | 17.8 / 7.03 | 17.90 / 7.29 | 18.12 / 7.38 | 17.98 / 8.10 |
| R/mR @50 | IMP | 18.2 / 7.05 | 17.53 / 6.10 | 17.74 / 6.43 | 16.89 / 5.11 |
| | MOTIFS | 21.7 / 9.57 | 18.26 / 7.03 | 18.38 / 7.12 | 17.89 / 6.70 |
| | VCTree | 22.1 / 10.2 | 17.94 / 6.84 | 18.47 / 7.44 | 17.97 / 6.52 |
| | GPS-Net | 19.6 / 7.49 | 19.44 / 7.77 | 19.78 / 7.85 | 19.36 / 8.46 |
| R/mR @100 | IMP | 18.6 / 7.23 | 18.09 / 6.28 | 18.20 / 6.58 | 17.26 / 5.22 |
| | MOTIFS | 22.0 / 9.69 | 18.75 / 7.23 | 18.88 / 7.23 | 18.39 / 6.83 |
| | VCTree | 22.5 / 10.2 | 18.52 / 7.00 | 18.92 / 7.56 | 18.47 / 6.64 |
| | GPS-Net | 20.1 / 7.67 | 19.89 / 7.86 | 20.14 / 7.94 | 19.83 / 8.62 |

**Benchmark setups:** To observe the effects of cluster imbalance, we do not equalize the number of data points. We tested 3 types of data partitioning as follows: **(1) Random**, where data is distributed randomly among all clients, ensuring nearly equal sizes for each. **(2) Shard-based partition IID**, where we set $p = 5$, where $p$ is the number of clusters that client sample from. As aforementioned, when $p$ equals the number of clusters, the data from each cluster is equally distributed among 100 clients, i.e., an IID case. **(3) Shard-based partition non-IID**, where we set $p = 1$ for imposing semantic heterogeneity. This time, clients were allocated to each cluster based on the amount of data in each cluster, rather than assigning an equal number of clients to all clusters. According to Table 5, [5, 58, 11, 7, 19] clients were assigned to each cluster, respectively, ensuring nearly equal sizes for each.

**Results:** The results in the Table 11 differ in some aspects from the analysis in the main paper. Firstly, the performance of all algorithms in FL scenarios has significantly increased. This is because the quantity of data assigned to each client has greatly increased and models are overfitted to the dominant cluster. The performance trends of each algorithm have changed as follows: In case of IMP Xu et al. (2017), the performance gap between CL and IID has significantly narrowed. While the performance in FL scenarios has greatly improved, the performance of CL has remained unchanged. This indicates that IMP is relatively less affected by the amount of data. And in case of MOTIFS Zellers et al. (2018) and VCTree Tang et al. (2019), the performance trends were almost similar to those observed in the previous experiments on balanced dataset. VCTree still appears to be slightly more vulnerable to data heterogeneity compared to MOTIFS. For IMP, MOTIFS and VCTree, it was observed that performance decreases as data heterogeneity increases. But for GPS-Net Lin et al. (2020), the trends observed were completely different from what was expected generally in FL scenarios. The performance reported in the previous study Yang et al. (2022) is lower than the performance on the balanced dataset. Previous research mentioned that the core of GPS-Net explicitly models the direction of predicates, which is why it does not perform well on PSG dataset. However, when examining the results in our main table, modeling the direction of predicates might not be the cause. Furthermore, GPS-Net showed the best results in the non-IID scenario. In fact, this is an unusual and greatly deviated from our expectations. Therefore, we concluded that GPS-Net performs well when clients have sufficient data and a relatively small number of categories. Additionally, we believe that a detailed performance comparison through FL on the SGG task, rather than PSG task, would allow for a more in-depth analysis.

### F.1 CONVERGENCE BEHAVIOR FOR IMBALANCE DATASET

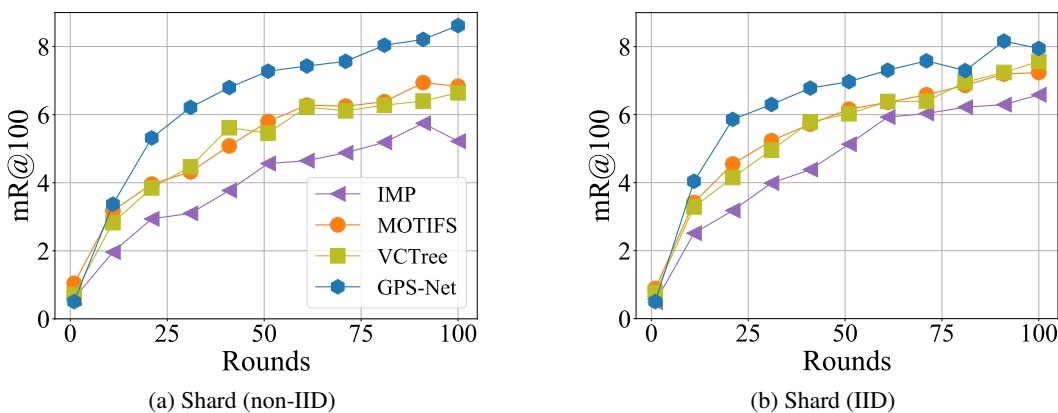

(a) Shard (non-IID)   (b) Shard (IID)

Figure 6: Convergence behaviors of the unbalanced dataset for IID and non-IID cases

We present the convergence behavior of four algorithms on the imbalance dataset for IID and non-IID case in Fig. 6. In contrast to the results from balanced datasets, there is almost no difference in convergence speed between IID and non-IID scenarios. This suggests that each client is sufficiently trained locally even in the non-IID condition, as they possess $4\times$ more data in the imbalanced dataset compared to the balanced one.

