# OpenReview forum: "Benchmarking Federated Learning for Semantic Datasets: Federated Scene Graph Generation"
_ICLR.cc/2025/Conference — ICLR 2025 Conference Withdrawn Submission_

### Official Review · Reviewer_sEoi · 2024-10-29

**Soundness:** 3
**Presentation:** 2
**Contribution:** 2
**Rating:** 3
**Confidence:** 4

**Summary:**

This paper applies FL to multi-semantic datasets and scene graph generation. It provides a benchmark for this new application with controllable semantic heterogeneity among clients. Experiemnts are conducted on FedAvg and FedAvgM algorithms with different PSG algorithms.

**Strengths:**

- This work proposes to apply FL into a new application, i.e. scene graph generation and multi-semantic settings. It is important to develop a benchmark for such a new application.
- Extensive experiments are conducted across multiple settings and algorithms.
- The organization of the paper is clear and easy to follow.

**Weaknesses:**

- It is good to apply FL to a new application, but the motivation is unclear. Why this application is important in FL?
- The writing of the paper could be further improved and it is generally not easy to follow. The structure of the paper could be improved. It would be better to introduce the main methodology / benchmark earlier.
- Various claims in the paper could be improved. For example,
    - 1. The paper emphaszes that existing works mostly focus on image classification, but there are many other applications of FL are studied, e.g., object detection, person ReID, face recognition, etc. it is important to discuss these tasks and justify how they are different from the proposed application.
    - 2. The author stated “there does not exist  a task-agnostic FL benchmark”. It is unclear what is the meaning of “task-agnostic” FL benchmark? Is the proposed benchmark a task-agnostic benchmark?
    - 3. The authors defined “FL testing environments”, but the mentioned works are FL benchmarks, libraries, or frameworks, it is uncommon to call them FL testing environments. Some related works are also not discussed, e.g., COALA
    - 4. “the most crucial factor in FL evaluation is demonstrating the effectiveness of methods with strong data heterogeneity across clients.”. There are many other aspects that are important, e.g., system heterogneneity, communication efficiency, etc.
- The paper uses numerous PSG algorithms without much explanation, why these algorithms are consider? why they are important?

**Questions:**

As stated in the Weakness section

---

### Official Review · Reviewer_uf5h · 2024-11-01

**Soundness:** 3
**Presentation:** 3
**Contribution:** 2
**Rating:** 5
**Confidence:** 3

**Summary:**

This paper proposes a novel methodology to create an FL benchmark for complex computer vision tasks that require reasoning about multiple objects and their semantics, such as panoptic scene graphs (PSG). The authors use this methodology to adapt the PSG dataset and benchmark various PSG methods with controllable semantic heterogeneity.

**Strengths:**

* The paper addresses an important gap in existing FL benchmarks, which primarily focus on simpler single-label tasks, by proposing a benchmark suited to more complex tasks that involve multiple semantics and inter-object relationships.

*  The authors propose a novel FL benchmark for the task of  Panoptic Scene Graph Generation.

* The paper is well structured and easy to read.

**Weaknesses:**

* Only one task is presented as a proof of concept, which is understandable; however, it raises questions about the methodology’s broader applicability, especially for tasks that integrate both vision and language in a VQA style. It would improve the paper if the authors could discuss the feasibility of extending this approach to other complex tasks with similar or higher levels of complexity. Comments in this direction would be greatly appreciated.

* The paper does not investigate the effects of using FL algorithms beyond FedAvg and FedAvgM, which limits insights into how the proposed methodology might perform with other FL approaches. Would it be possible for the authors to comment on this and expand into why  those two algorithms were the ones  evaluated?

* While I appreciate the discussion on the methodology to create the benchmark and find it  interesting, the paper could benefit from further discussion on practical insights derived from using this benchmark. For example, summarizing key trade-offs that practitioners might consider when applying FedAvg to complex visual tasks with multi-semantic labels could add value.

**Questions:**

see above

---

### Official Review · Reviewer_fq8Y · 2024-11-02

**Soundness:** 3
**Presentation:** 3
**Contribution:** 2
**Rating:** 3
**Confidence:** 4

**Summary:**

This paper mainly focuses on the datasets used in federated learning, arguing that the datasets traditionally employed in federated learning, which are primarily based on classification and consist of samples paired with single category labels, are no longer suitable for current deep learning tasks that involve understanding the deeper semantic information in samples, such as multiple objects and their relationships. Additionally, this paper highlights that the past methods considered only the heterogeneity of label distributions among clients, lacking attention to semantic heterogeneity. To address these issues, this paper utilizes the PSG dataset, which includes scene graphs, summarizes the scene graphs into category tensors, and clusters images based on these category tensors, resulting in five major categories with different semantics that are assigned to different clients. This paper then test the performance of centralized training against various federated learning methods on the formed dataset.

**Strengths:**

1. The method for constructing semantic heterogeneity among clients is novel.
2. This paper has clear logic and easy to understand.

**Weaknesses:**

1. The complex tasks proposed in the paper, which require deep semantic information from samples, demand large-scale data. However, the paper also requires the dataset to provide scene graph information, which incurs substantial annotation costs. For reference, most multimodal large-scale models that handle these complex tasks are trained on unlabeled data.
2. The analysis presented in the paper is based solely on the PSG dataset, and it remains unclear whether similar clustering performance applying to other datasets.
3. The paper does not provide a unique client partitioning method to reflect semantic heterogeneity. The partitioning methods in Section 4.2 are commonly used in other federated learning methods.
4. The method relies on the used scene graph generation method and requires prior knowledge of the correspondence between objects and their categories.

**Questions:**

See weaknesses.

---

### Official Review · Reviewer_6zPK · 2024-11-05

**Soundness:** 2
**Presentation:** 2
**Contribution:** 2
**Rating:** 3
**Confidence:** 5

**Summary:**

This paper focuses on using federated learning (FL) to address Scene Graph Generation (SGG) and Panoptic Scene Graph Generation (PSG) problems. It introduces a federated learning benchmark designed to create an FL benchmark with controlled cross-client semantic heterogeneity. Through data clustering and heterogeneous data distribution, the method addresses complex semantic challenges in FL contexts. Experimental results demonstrate the effectiveness of the proposed approach.

**Strengths:**

1. This paper applies the federated learning framework to the SGG/PSG problem, which is more challenging than the ordinary classification problem.

2. This paper focuses on the data heterogeneity problem in federated learning and constructs a client-side data partitioning approach in federated learning from the perspective of multi-semantic and multi-labeling. These works contribute to the later application of federated learning to complex semantic scenarios.

**Weaknesses:**

1. There are problems with the presentation of certain words in the writing of this paper. For example, the first sentence of the abstract, federated learning is a distributed machine learning paradigm, but it can not be directly said that it is decentralized, but also includes centralized, and the method of this paper is based on centralized federated learning.

2. The abstract part, less description of their own methods and innovations, and a large description of background knowledge. Through the abstract, it is impossible to understand the key innovation of this article.

3. Most of the references involved in this article are from 2022 and before, and the latest related work is described too little, and it is hoped that the authors can conduct more extensive research.

4. In terms of data division, data clustering is targeted to solve the problem of mult-isemantics. However, the later data partitioning, block partitioning and Dirichlet-based partitioning seem to be the migration of the existing data partitioning methods in federated learning, and the innovation is limited.

5. Section 5.3 of the experimental part of the FedAvgM method, why does it improve the effect, what design makes its improve the effect? These analyses should be added.

6. As a benchmark framework, it should be extensively validated on more datasets and thus be applicable to more scenarios.

**Questions:**

1. Starting from the introduction, the authors have been emphasizing that their innovation is the first time they have proposed a solution to the multisemantic problem in a federated learning scenario. This innovation is insufficient, and the method section does not see the authors propose any coping strategies for the problem of semantic heterogeneity, such as collaboration methods between clients, server aggregation methods, and so on.

2. In the related work section, in the first paragraph, the authors list many federated learning methods. What is the relationship between these methods and this paper? Why are they listed here? A simple stack of related work is not the best way to present it.

3. what are some of the FL benchmarks that are already available and what are the advantages and disadvantages of their various? What are the applicable scenarios? These need to be introduced specifically.

4. How does the framework of federated learning work for problems like SGG/PSG, and what is the whole process like? Including training and testing. Involving the selection of clients, the aggregation of servers should be expressed clearly or reflected in the framework diagram.

5. Is there a detailed description of the work “IMP, MOTIFS, VCTree, GPS-Net” compared in the experimental part?

---

### Note · Authors · 2024-11-15

**Comment:**

Dear Reviewers,

Thank you for taking the time to review our paper and providing valuable feedback. I appreciate your efforts and insights, which have highlighted important points for improvement. Although We have decided to withdraw the paper at this time, your comments have given me a clearer direction for future work, and I look forward to applying your suggestions to enhance my research.

Thank you once again for your thoughtful and constructive feedback.

Best regards,
Authors.

**Withdrawal Confirmation:**

I have read and agree with the venue's withdrawal policy on behalf of myself and my co-authors.